# OpenReview forum: "Abstractive Red-Teaming of Language Model Character"
_ICLR.cc/2026/Conference — ICLR 2026 Conference Withdrawn Submission_

### Official Review · Reviewer_adWE · 2025-10-27

**Soundness:** 2
**Presentation:** 2
**Contribution:** 2
**Rating:** 2
**Confidence:** 4

**Summary:**

The paper presents a new red-teaming/jailbreaking approach using a bilevel setup. In the first step, the framework generates high-level categories of questions that could elicit incorrect behavior from a target LLM. In the next step, queries are generated based on the categories in the previous step, which are finally fed to the LLM. The paper proposes two approaches to optimize this framework — one using the leave-one-out REINFORCE algorithm, where the reward is based on whether the LLM is jailbroken, and a second approach uses Monte Carlo estimates to choose a category. The paper conducts experiments on a large number of LLMs and reports the results of each approach.

**Strengths:**

**Strengths**:

1. The paper presents a natural way of decomposing the red-teaming problem in a bilevel setup. Intuitively, it is easier to search over the categories of problematic queries rather than the query space itself.
2. The paper proposes two different approaches to optimize the above framework. The first approach trains a category generator using reinforcement learning, where the reward is a binary label indicating whether the LLM was jailbroken. The second one searches over the category space and chooses categories based on the jailbreaking scores obtained using Monte-carlo.
3. The paper provides extensive details on data collection and reward model training to implement the above framework. The paper also presents a range of experiments on a large number of LLMs.

**Weaknesses:**

**Weaknesses**:

1. The paper doesn’t present any baselines apart from a random sampling. There are a large number of jailbreaking papers; please refer to [1] for an incomplete list. The paper should compare with the state-of-the-art methods to establish the impact of their proposed method.
2. The paper doesn’t use standard metrics to evaluate its approach. The paper reports the mean rewards obtained using the reward model, but the details of the scoring mechanism aren’t described in the main body of the paper. The paper should report standard metrics like success rate in jailbreaking LLMs. This method should also be evaluated against state-of-the-art LLMs like GPT-5 and Gemini. This would demonstrate the effectiveness of the method in scalable systems.

Comments:

Line 168: Minor — the set of all possible strings is finite, therefore the mentioned set is also finite.

[1] A Comprehensive Study of Jailbreak Attack versus Defense for Large Language Models, Xu et al., 2024

**Questions:**

Please respond to the weaknesses and comments in the above section.

---

### Official Review · Reviewer_7tFL · 2025-11-01

**Soundness:** 2
**Presentation:** 2
**Contribution:** 2
**Rating:** 4
**Confidence:** 4

**Summary:**

This paper introduces Abstractive Red Teaming (ART), a framework for automatically generating adversarial prompts by abstracting harmful queries through large language models (LLMs). The goal is to generate semantically equivalent but surface-different prompts that bypass safety filters. The authors propose an iterative paraphrasing and abstraction process, guided by a “semantic preservation” objective and a toxicity classifier. ART aims to generate diverse adversarial prompts that remain challenging for LLM safety systems to detect.

**Strengths:**

1. Targets a relevant and practical problem aobut automating adversarial prompt discovery.
2. Provides clear examples and evaluation results that are easy to reproduce.
3. Attempts to move beyond surface-level perturbations by introducing a “semantic abstraction” step.

**Weaknesses:**

1. Low Novelty and Conceptual Incrementality. The proposed approach primarily reformulates existing red teaming and paraphrasing strategies under a new expression. While the authors frame this as an innovative abstraction-driven method, the actual process about iterative rewording and filtering based on similarity or toxicity classifiers is a minor variation of well-known paraphrase-based adversarial generation.
2. Template Dependence and Limited Diversity. The system’s performance appears heavily tied to the chosen abstraction templates and instruction patterns used during generation. In practice, the success of the attacks relies on specific prompt wordings and fixed structural cues, which makes the method brittle and less generalizable. When applied to different safety domains or LLM families with distinct guardrail training, the approach may fail to adapt.
3. Weak Signal Design and Limited Generalization. The “semantic preservation” and “toxicity-guided” objectives used to control abstraction quality are simplistic and loosely connected to actual adversarial effectiveness. Embedding similarity or classifier confidence cannot reliably ensure semantic equivalence or adversarial strength, especially in nuanced safety contexts. As a result, the method risks generating either trivial rephrasings or semantically drifted prompts that no longer test model robustness meaningfully.

**Questions:**

1. How sensitive are results to the chosen abstraction templates and initial seed prompts?

---

### Official Review · Reviewer_kSd8 · 2025-11-01

**Soundness:** 3
**Presentation:** 3
**Contribution:** 3
**Rating:** 6
**Confidence:** 3

**Summary:**

This paper proposes to improve red teaming strategy from individual adversarial prompts to query categories, which is represented as natural language descriptions of classes of user queries that may trigger policy violations. This paper introduces two algorithms: CRL (Category-Level RL), which applies REINFORCE optimization directly in categories space, and QCI (Query–Category Iteration), which alternates between exploration (generating high-risk query examples) and exploitation (summarizing them into new categories).
Experiments on 4 LLMs show that CRL and QCI outperform random sampling in discovering realistic and high-risk scenarios.

**Strengths:**

1. Moving from query-level to category-level search is conceptually reasonable and practically relevant.
2. The two algorithms (CRL and QCI) are well structured.
3. Rich qualitative findings: The paper provides many concrete and interesting categories as cases, demonstrating the effectiveness of the proposed method.

**Weaknesses:**

1. The comparison is mainly against random sampling; this paper should have more competitive baselines (e.g., some taxonomy-guided methods).
2. The total query budget is large (100k queries for each model, CRL seems to require large amount of queries), and there’s limited discussion of convergence, sample efficiency, or sensitivity to hyperparameters.

**Questions:**

See weaknesses, and

1. How many preferences data were used to train the reward model?
2. How did you address the gap from generated to real user data in discovered categories?

---

### Official Review · Reviewer_cSP7 · 2025-11-03

**Soundness:** 2
**Presentation:** 2
**Contribution:** 3
**Rating:** 4
**Confidence:** 4

**Summary:**

This paper presents an interesting approach to uncovering the types of natural queries that cause language models to violate their intended character specifications. Instead of focusing on individual adversarial prompts, it proposes optimizing a category generator via REINFORCE-based reinforcement learning to iteratively discover high-risk categories of harmful queries. The results suggest that different models exhibit distinct vulnerability patterns across query categories, highlighting model-specific behavioral weaknesses.

**Strengths:**

1.	Understanding what kinds of user queries lead to policy violations is crucial for improving the safety of LLMs in real-world applications.
2.	Shifting focus from specific adversarial queries to learned semantic categories offers a different way to analyze model vulnerabilities - this conceptual move from instances to abstractions is valuable.

**Weaknesses:**

1.	Overall, the current presentation lacks clarity and structure. A visual illustration (e.g., a diagram showing the interaction between the category generator, reward model, and experience pool) would greatly aid understanding. Additionally:
- The core algorithm should be moved from the appendix into the main text.
- The term “a subset of query space” (Section 4.2) is used without formal definition—is this a set of strings, embeddings, or structured attributes?
- What is the format of the generated categories? Are they free-text descriptions, templates, or structured labels?
- Lines 262–269: Why sample K/(ℓ+1) subsets? This choice seems arbitrary without justification.
- It’s unclear which component drives diversity and novelty in the discovered categories. Is it the sampling strategy, the reward signal, or the generator itself?
2.	In Section 4.1.2, why not use off-the-shelf safety classifiers such as Llama Guard or QwenGuard as the reward model? If a custom reward model is preferred, how does its training data relate to the domains and categories present in the experience pool? Any overlap could introduce circularity or overestimation of risk.
3.	What if one directly optimizes over individual queries instead of categories? This would help isolate the benefit of the categorization framework. Besides, the paper should compare against recent red-teaming and failure-exploration methods, such as:

[1] Diverse and Effective Red Teaming with Auto-generated Rewards and Multi-step RL

[2] Forewarned is Forearmed: Leveraging LLMs for Data Synthesis through Failure-Inducing Exploration

[3] LLMs Know Their Vulnerabilities: Uncover Safety Gaps through Natural Distribution Shift  and so on.

4.	The experimental setup uses different settings for Llama-3.1-8B-Instruct versus other models. Could the authors provide more justification for these variations?
4.	While the qualitative analysis in Section 4.3 is insightful and engaging, many figures and tables are placed to the appendix. Given their importance for illustrating the types of discovered categories, key examples should be included in the main paper to improve readability and impact.

**Questions:**

please refer to the above Weaknesses

---

### Note · Authors · 2025-11-30

**Comment:**

We thank the reviewers for their time and comments, which will aid in improving our paper.

**Withdrawal Confirmation:**

I have read and agree with the venue's withdrawal policy on behalf of myself and my co-authors.